# Simulated coordinated impacts of the previous autumn NAO and winter El Niño on the winter aerosol concentrations over eastern China

Juan Feng[1], Jianping Li[2,1], Hong Liao[3], and Jianlei Zhu[4]

1. *College of Global Change and Earth System Science, Beijing Normal University, Beijing, China*

2. *Key Laboratory of Physical Oceanography–Institute for Advanced Ocean Studies, Ocean University of China and Qingdao National Laboratory for Marine Science and Technology, Qingdao 266003, China*

3. *School of Environmental Science and Engineering, Nanjing University of Information Science & Technology, Nanjing, China*

4. *China-ASEAN Environmental Cooperation Center, Beijing, China*

***Corresponding author*:**

*Dr. Juan Feng*

*College of Global Change and Earth System Science (GCESS),*

*Beijing Normal University, Beijing 100875, China*

Tel: 86-10-58802762

Email: fengjuan@bnu.edu.cn

# Abstract

The high aerosol concentrations (AC) over eastern China have attracted attention from both science and society. Based on the simulations of a chemical transport model using a fixed emissions level, the possible role of the previous autumn North Atlantic Oscillation (NAO) combined with the simultaneous El Niño-South Oscillation (ENSO) on the boreal winter AC over eastern China is investigated. We find that the NAO only manifests its negative impacts on the AC during its negative phase over central China, and a significant positive influence on the distribution of AC is observed over south China only during the warm events of ENSO. The impact of the previous NAO on the AC occurs via an anomalous sea surface temperature tripole pattern by which a teleconnection wave train is induced that results in anomalous convergence over central China. In contrast, the occurrence of ENSO events may induce an anomalous shift in the western Pacific subtropical high and result in anomalous southwesterlies over south China. The anomalous circulations associated with a negative NAO and El Niño are not favorable for the transport of AC and correspond to worsening air conditions over central and south China. The results highlight that the combined effects of tropical and extratropical systems play considerable role in affecting the boreal winter AC over eastern China.

## 1. Introduction

Atmospheric particles (i.e., aerosols) are the key pollutants that exhibit an important adverse impact on human health, environmental pollution, global climate change, and atmospheric visibility (IPCC, 2013). Aerosol particles may alter the precipitation rates and optical properties of clouds (Hansen et al., 1997), impacting the radiation balance of the entire Earth-atmosphere system via absorbing and scattering solar radiation (Jiang et al., 2017; Yue and Unger, 2017). A better understanding of aerosol variations is therefore important and useful for scientific and social endeavors.

The meteorology parameters, i.e., atmospheric temperature (Aw and Kleeman, 2003; Liao et al., 2015), boundary layer (Kleeman, 2008; Yang et al., 2016), wind (Zhu et al., 2012; Yang et al., 2014, 2017; Feng et al., 2017), and humidity (Ding and Liu, 2014), show a non-negligible impact on the regional aerosol concentrations (AC) via affecting the deposition and transportation processions. Moreover, the intraseasonal and interannual variations in climatic phenomena could affect both the spatial and temporal accumulation and distribution of AC due to the associated variations in the circulation and rainfall anomalies. For example, the monsoon onset could affect the seasonal variations in regional AC (Tan et al., 1998; Chen and Yang, 2008). The interannual variation of AC over East Asia is connected with the interannual variation of East Asian winter monsoon (Jeong and Park, 2016; Lou et al., 2016, 2018; Mao et al., 2017) and summer monsoon (EASM; Zhang et al., 2010; Zhu et al., 2012). The seasonal evolution of the El Niño-South Oscillation (ENSO) impacts the seasonal variations of AC over northern and southern China (Liu et al., 2013; Feng et al., 2016a, 2017). The AC

variation in the US is influenced by the Pacific Decadal Oscillation (Singh and
Palazoglu, 2012). These findings suggest that the role of climate systems in impacting
the regional air quality cannot be ignored.
The North Atlantic Oscillation (NAO), reflecting large scale fluctuations in
pressure between the subpolar low and subtropical high, is one of the most determinant
and influential climate variability modes in the extratropical Atlantic Ocean, (e.g.,
Hurrell, 1995; Gong et al., 2001; Visbeck et al., 2001). A negative (positive) polarity of
the NAO is reflected by positive (negative) pressure anomalies over the high latitudes
of the North Atlantic and negative (positive) pressure anomalies over the central North
Atlantic. Both the positive and negative phases of NAO are accompanied with large
scale modulations in the location and intensity of the North Atlantic jet stream and
storm track (Gong et al., 2001; Li and Wang, 2003). The surface layer wind would vary
associated with changes in the jet stream because of the NAO's quasi-barotropic
characteristic, resulting in varied Ekman heat transport and basin-wide variations in the
underlying sea surface temperatures (SST; Marshall et al., 2001; Wu et al., 2009; Wu
and Wu, 2018).
The NAO massively impacts the temperature and precipitation patterns over the
US and central Europe, i.e., a wet and warm winter in Europe, and mild and wet winter
conditions would be expected accompanied with a positive NAO phase. Moreover, the
NAO exhibits significant cross-seasonal impacts on the downstream regional climate.
For example, it is reported that variation in boreal spring NAO influenced the
subsequent intensity of the EASM from 1979-2006 (Wu et al., 2009). The linkage
between the EASM and NAO has been further explored but on the interdecadal scale
(Wu and Lin, 2012; Wu et al., 2012; Zuo et al., 2013), and it is suggested that the
preceding spring NAO dominated the relationship of the NAO-EASM more than the
simultaneous summer NAO, similar result is seen in Zheng et al. (2016). Xu et al. (2013)
presented that the previous boreal summer NAO significantly influenced the following
September rainfall over central China. These studies highlight the important role of the
NAO signal on the climate in East Asia, especially the cross-seasonal impacts, which
are beneficial for seasonal forecasting.

In addition to the influence of the extratropics, the impact originating from the

tropics is another important driver of the climate anomalies in China. As the most
dominant interannual variability of the tropical air-sea coupled system, the El Niño-
Southern Oscillation (ENSO) exhibits profound influences on the weather and climate
around the world (e.g., Ropelewski and Halpert, 1987; Harrison and Larkin, 1998). The
occurrence of ENSO phenomenon displays significant effects in impacting the global
and regional oceanic and atmospheric anomalous patterns (e.g., Rasmusson and
Carpenter, 1982; Trenberth, 1997). The seasonal climate variation in China is closely
linked with the evolution of ENSO events. For example, increased rainfall is expected
to be found over the Huai-he and Yangtze River valley, whereas less rainfall is seen
over northern and southern China during the decaying summer of an El Niño event
(Zhang et al., 1996, 1999; Ye and Wu, 2018). During the developing autumn of an El
Niño event, enhanced rainfall would be expected over southern China due to the
associated anomalous shift in the western Pacific subtropical high (WPSH). However,
without significant influence during the developing summer (Feng et al., 2016b).
During the mature winter, both the warm and cold events show significant impacts on
the temperature and rainfall anomalies over eastern China (Weng et al., 2009; Wu et al.,
2011; Wu and Zhang, 2015; Li et al., 2019; Zhang et al., 2019a, 2019b).
As shown above, both the NAO and ENSO significantly impact the climate over
China. China now suffering from relatively high aerosol loading, and this is commonly
ascribed to the increased emissions connected with the speedy economic growth.
However, as discussed above that the role of meteorological conditions in affecting the
AC cannot be ignored. Accordingly, it is of interest to explore the possible impacts of
the NAO and ENSO on the distributions of AC over China. The possible impacts of the
NAO on the aerosol has been discussed by Moulin et al. (1997) and Jerez et al. (2013);
however, they concentrated on its influences on the North Atlantic Ocean and Europe,
respectively. Feng et al. (2016a) indicated the potential effects of El Niño on the AC
over China, but with a focus on the seasonal evolution. Therefore, does the NAO exhibit
significant impacts on the AC, and how the combination of the NAO and ENSO affect
the distribution of AC over China, as both of them show important modulation of the
climate over China.
The above discussions provide the main motivation of the present work. The
conditions in boreal winter are discussed in the present work, as this time is
corresponding to the heat supply season and the AC over China peak during this season.
The coordinated role of the previous autumn (September to November, SON) NAO and
the simultaneous ENSO is compared to that of the NAO alone, and also as well as the
involved physical mechanisms. The rest of this paper is arranged as follows. Model,
datasets, and methodology employed are presented in Section 2. The possible impacts
of the NAO and ENSO on the AC are explored in Section 3. Section 4 discusses the
involved physical mechanism. Section 5 provides the discussion and conclusions.

## 136      2. Datasets, simulations, and methodology

### 137      2.1 Datasets

The input background meteorological variables of the GEOS-Chem model show
high degree of uniformity with the current widely used reanalyses (e.g., Zhu et al., 2012;
Yang et al., 2014). Here, the SLP in the National Centers for Environmental
Prediction/National Center for Atmospheric Research (NCEP/NCAR) reanalysis
(Kalnay et al., 1996) with a 2.5° latitude × 2.5° longitude resolution, and the UK
Meteorological Office Hadley Centre's sea ice and SST datasets (HadISST; Rayner et
al., 2003) with a 1° latitude × 1° longitude resolution are used to verify the reliability
of the Goddard Earth Observing System, Version 4 (GEOS-4).

### 146      2.2 GEOS-Chem simulations

The influences of the NAO on the simulated AC over China are examined using a
three-dimensional tropospheric chemistry model, i.e., GEOS-Chem (version 8.02.01;
Bey et al., 2001). The model is driven by assimilated meteorological fields from the
GEOS-4 of the NASA Global Modeling and Assimilation Office, with a 2° latitude ×
2.5° longitude resolution, and 30 hybrid vertical levels. This model contains a detailed
coupled treatment of tropospheric ozone-NOx-hydrocarbon chemistry, as well as

aerosols and their precursors, containing nitrate, black carbon, sulfate, sea salt, ammonium, mineral dust, dust aerosols, and organic carbon (Bey et al., 2001; Liao et al., 2007). The aerosol dry and wet depositions follow Wesely (1989) and Liu et al. (2001), with details in Wang et al. (1998). According to Liao et al. (2007), the AC were defined as PM2.5 as follows,

$$[PM_{2.5}] = 1.37 \times [SO_4^{2-}] + 1.29 \times [NO_3^-] + [POA] + [BC] + [SOA] \quad (1)$$

$SO_4^{2-}$, $NO_3^-$, POA, BC, and SOA are the aerosols particles of sulfate, nitrate, primary organic aerosol, black carbon, and second organic aerosol, respectively. The sea salt aerosols and mineral dust are not considered for that measurements indicate that they are not the major aerosol species in the eastern China during winter (Xuan et al., 2000; Duan et al., 2006).

The anthropogenic emissions in the GEOS-Chem and experiment design are similar to Zhu et al. (2012), in which the biomass burning emissions and anthropogenic emissions are fixed at year 2005 level in the simulation. That is the observed variations in the distributions of AC as seen below was due to the variations in meteorological conditions associated with climate events. Due to the longevity of the GEOS-4 datasets, the period 1986-2006 is focused on. GEOS-Chem is a well-recognized atmospheric chemistry model and is widely utilized due to its capability to well characterize the seasonal, interannual, and decadal variations of pollutant aerosols in the East Asia and beyond (e.g., Zhu et al., 2012; Yang et al., 2014, 2016; Feng et al., 2017). The well performance and wide application of GEOS-Chem provide confidence for employing

the model to investigate the coordinated impacts of NAO and El Niño on the AC over
eastern China.
**2.3 NAO index and Niño3 index**
The NAO index (NAOI) is employed to quantify the variations in the NAO phase
(Hurrel et al., 1995; Gong and Wang, 2001). The definition of the NAOI follows Li and
Wang (2003) and is calculated as the zonal mean SLP difference between 35°N (i.e.,
refers to the mid-latitude center) and 65°N (i.e., refers to the high latitude center) from
80°W to 30°E over the North Atlantic by
$$\mathrm{NAOI} = \hat{P}_{35°N} - \hat{P}_{65°N} \tag{2}$$
where $P$ is the monthly mean SLP averaged from 80°W to 30°E, $\hat{P}$ is the normalized
value of $P$, and the subscripts indicate latitudes. For a given month $m$ in year $n$, the
normalization $\hat{P}$ is defined as follows
$$\hat{P}_{n,m} = \frac{P'_{n,m}}{S_P} \tag{3}$$
where $P'_{n,m}$ is the monthly pressure anomaly of $P_{n,m}$, departure from period 1986-
2006, and $S_P$ is the total standard deviation of the monthly anomaly $P'_{n,m}$,
$$S_P = \sqrt{\frac{1}{12 \times 21} \sum_{i=1986}^{2006} \sum_{j=1}^{12} P'^2_{j,i}} \tag{4}$$
The monthly NAOI is calculated based on the monthly mean SLP from both the
NCEP/NCAR and GEOS-4 assimilated meteorological dataset for 1986-2006. The
boreal autumn NAOI is defined as the average of the monthly NAOI during September,
October, and November (Fig. 1). The series of NAOI show strong interannual variations,
and the two series based on GEOS-4 and NCEP/NCAR are closely correlated with each
other with a significant coefficient of 0.98, implying the GEOS-4 dataset could capture
the variation in the NAO.

El Niño events were defined as standardized 3-month running mean Niño3 index

(areal mean SST averaged over 150°-90°W, 5°N-5°S) above 0.5°C and persisting for at
least 6 months. The skin temperature (i.e., SST over ocean and surface air temperature
on land) was employed to obtain the Niño3 index for that SST is not available in the
GEOS-4 meteorological dataset. The boreal winter Niño3 index is calculated as the
average of the monthly Niño3 during December, January, and February, i.e., winter
1997 is for the December 1997 and January and February 1998. The boreal winter
Niño3 indices based on the GEOS-4 and HadISST are significantly correlated with each
other, (Fig. 1), with a coefficient of 0.99. The high correlations among the indices
further indicate the reliability of the model data.

## 3.  Influences of the NAO and El Niño on the AC over China

### 3.1  Climatological Characteristics of the AC

The spatial distribution of the standard deviation of boreal winter AC is shown in

Fig. 2. Eastern China (105°E eastward, 35°N southward) shows high loading of
aerosols in both the column and surface layer concentrations (figure not shown). Further,
the variance of winter AC over eastern China is most pronounced compared to other
regions during this season (Fig. 2a, b). As an evident monsoonal region, eastern Asia is
influenced by winter monsoon, i.e., a strong Aleutian low is seen in the north Pacific,
and the Asian continent is controlled by the Siberian high during boreal winter. The
strong pressure gradient between the Siberian high and Aleutian low results in strong
northwesterlies prevailing over eastern China (Fig. 2c).

## 3.2 Relationships between the AC & NAO and El Niño

The spatial distribution between the surface AC and previous autumn NAOI and
simultaneous winter Niño3 index are presented in Fig. 3. Positive correlations are seen
over south (30°N south) and northwest China in the correlations with the Niño3 index,
indicating that a warm ENSO event would associate with high AC over south and
northwest China. In contrast, negative correlations over south and central China are
observed in the correlations with autumn NAO, implying a positive NAO phase is
linked with less AC over these regions, thus favoring better air conditions. The analysis
suggests that the ENSO and NAO show opposite effects on AC over south China, i.e.,
the NAO displays a negative impact and the ENSO displays a positive impact. However,
the relationship between the autumn NAOI and winter Niño3 index is insignificant with
a correlation of -0.08 during period 1986-2006.
The above relationships are further examined in their positive and negative phases,
as strong asymmetry was reported in the climatic impacts of the NAO (Xu et al., 2013;
Zhang et al., 2015) and ENSO (Cai and Cowan, 2009; Karior et al., 2013; Feng et al.,
2016b). The asymmetric influences of the NAO and ENSO on AC are obvious in the
spatial distributions of the linear correlation coefficients (Fig. 4). During the El Niño
events, south China is impacted by significant positive correlations, in contrast, a non-

significant correlation is observed over this region during the La Niña events. This point

implies the significant relationships between the ENSO and AC over south China are

mainly connected with warm events, i.e., El Niño. The negative correlations between

the NAO and AC mainly occurred in the negative phase of the NAO, and the significant

correlations are mainly located in central China (lie from 28°N to 40°N). Thus, the

ENSO affects the distribution of AC in south China, but the impact is manifested during

warm events. Similarly, the effect of the NAO on the distribution of AC over central

China is only apparent during its negative phase.

  The results suggest that if the occurrence of a negative polarity of NAO overlaps

with an El Niño event, the combined effects of the two may further worsen the AC over

eastern China. In contrast, a solo occurrence of a negative NAO event is associated with

above-normal AC over central China. The statistic significant impacts of the negative

NAO and El Niño events on the AC could be further established by case study. Two

cases, i.e., the co-occurrence of an El Niño event and a negative NAO, and a solo

negative NAO event, were chosen to further explore the effect of the NAO and El Niño

on the AC over China. From 1986-2006, there are two years (1997 and 2002) with

equivalent negative values of autumn NAOI (-1.507 in 1997, and -1.510 in 2002).

Winter 1997 corresponds with the strongest El Niño in the past 120 years and winter

2002 corresponds with a neutral ENSO event. Consequently, the anomalous distribution

of AC during these two years are discussed in the context of comparing the combined

and solo effects of a negative NAO and El Niño in impacting the distribution of AC

over eastern China.

**3.3 Influences of the NAO & El Niño vs. the NAO on the AC**

Figure 5 presents the layer and column AC anomalies simulated for the winters of 1997 and 2002 departure from the climatological mean. Under the combined influence of a negative NAO and El Niño (1997), positive aerosol concentration anomalies are observed over eastern China (Fig. 5a, c). In addition, simulated enhanced AC were observed over central China in winter 2002 under the impacts of a negative NAO (Fig. 5b, d). These characteristics are also apparent in the vertical distribution (Fig. 6), which shows the zonal mean anomalies averaged over eastern China (105°–120°E). For winter 1997, increased AC cover the whole eastern China, with maximum values approximately 30°N, where the effects of the NAO and El Niño overlap (Figs. 4a, d). The combined effects of the anomalies show a consistent distribution in the vertical levels (Fig. 6). In contrast, evident increased AC anomalies are seen in central China, with the maximum at approximately 32°N during winter 2002.

The consistent results between the correlations and anomalies during the two cases highlight the role of the negative NAO and El Niño events in determining the distribution of AC over eastern China. The NAO shows a significant influence on the central China AC that are only apparent during its negative phase, and the ENSO impacts the AC over south China mainly during warm events.

# 4. Mechanisms of the effects of the NAO and El Niño on the AC

**4.1 Role of circulation transport**

The corresponding reverse role of the NAO and El Niño in impacting the
distribution of AC is mainly derived from their contrasting effects on circulation. Figure
7 shows the SLP and surface wind anomalies during the autumns of 1997 and 2002,
presenting an anomalously weak autumn NAO pattern. The negative phase of the NAO
displays as an anomalous SLP dipole structure between the middle latitude North
Atlantic Ocean and Arctic, i.e., with positive SLP anomalies at the Arctic over the
Atlantic sector, and anomalous negative SLP at middle latitude. Although the locations
of the anomalous pressure centers in the two negative NAO events show difference, the
anomalous SLP amplitude in the two events are similar, i.e., with greater negative SLP
anomalies at mid-latitudes, indicating that the pressure gradient of the two NAO
negative events is similar. The oscillation in the SLP is connected with anomalies in the
surface wind across the North Atlantic, i.e., associated with an anomalous cyclonic
centered approximately 45°N and anti-cyclonic circulation anomalies around Iceland.
During boreal winter and spring, an anomalous NAO could result in a tripole SST
anomalous pattern in the North Atlantic Ocean (Watanabe et al., 1999). A similar SST
tripole pattern is observed during boreal autumn, with warm SST anomalies at high and
low latitudes, and negative SST anomalies at middle latitudes in the North Atlantic
sector (Fig. 8a, c). Note that the negative SST anomalies during 1997 displays an east-
west direction but originated from a northwest-southeast direction during 2002 due to
the different locations of anomalous SLP (Fig. 7).
The North Atlantic anomalous SST tripole pattern is due to the feedback between
wind-SST, i.e., the anomalous anti-cyclonic (cyclonic) circulation weaken (strengthens)

the prevailing westerlies, which would result in decreased (increased) loss of heat and

warmer (cooler) anomalies in Ekman heat transport (Xie, 2004; Wu et al., 2009), and

is connected to warmer (cooler) local SST. Due to the short memory of the atmosphere,

the cross-seasonal influences of the NAO on the AC should be preserved in the

boundary layer forcing such as SST (Charney and Shukla, 1981). This anomalous

tripole SST pattern could persist to the following winter (Fig. 8b, d), as the anomalous

tripole SST pattern during winter and autumn show high consistencies in both 1997 and

2002, with significant spatial correlation coefficients of 0.32 and 0.51 between the

autumn and winter tripole SST patterns for 1997 and 2002, respectively.

Figure 9 shows the anomalous divergence at the upper troposphere. The

occurrence of a negative NAO phase is accompanied by an anomalous teleconnection

wave train over northern Eurasia (AEA) in the upper troposphere during boreal summer

(Li and Ruan, 2018). This anomalous teleconnection pattern is also observed during

boreal winter, with a shift in the precise locations. Under the influence of the anomalous

downstream teleconnection, north China is influenced by convergence anomalies, with

the center positioned over central China (Fig. 9). The anomalous convergence is clearly

seen in both the upper and lower troposphere, accompanied by anomalous easterlies or

southeasterlies over central China (Fig. 10). The direction of the anomalous wind is

opposite to the climatological winds, which would weaken the climatological wind and

is unfavorable for the transport of aerosol concentration, leading to increased AC over

central China, as displayed in Fig. 5.

For the winter 1997, corresponding to the El Niño's mature phase, south China
was influenced by an evident anomalous divergence at the lower troposphere,
indicating anomalous anticyclonic circulation over the coastal regions (Fig. 10a).
Anomalous southwesterlies prevailed in south China, implying weakened northerlies.
That is the anomalous meteorological conditions are unfavorable for aerosols transport
in the region and would result in a worsen air quality. In contrast, for the winter 2002,
south China was controlled by an anomalous divergence for that the main body of the
WPSH shifts to the south of south China (Fig. 10b). The anomalous circulation was
favorable for the emission of pollutant. Moreover, an evident anomalous divergence
was observed in south China in the winters of 1997 and 2002 at the upper troposphere;
however, the corresponding distribution of AC over this region is different. This
highlights the role of El Niño in impacting the circulation anomalies over south China,
as mentioned above. The occurrence of El Niño events would be accompanied by a
northwest shift of the WPSH during boreal winter and enhanced southwesterlies over
south China (Weng et al., 2009). Besides, column AC are mainly contributed by
concentrations at lower troposphere, suggesting that the lower troposphere circulation
may play a vital role in impacting the AC over south China.
**4.2  Role of wet deposit**
In addition to the contribution of the circulation anomalies to the distribution of
AC, changes in wet deposit also could affect distribution of AC. Figure 11 presents the
simulated wet deposit anomalies during the winters of 1997 and 2002. Negative
anomalies occurred over eastern China during the winter of 1997, favorable for
increased AC. This suggests the wet deposit plays a positive role in the enhanced AC
during winter 1997. Positive anomalies were observed over central China in the 2002
winter, inconsistent with the AC anomalies. The anomalous wet deposit during winter
of 1997 is paralleling to the AC anomalies over eastern China; however, not consistent
with that for the winter of 2002. This suggests that role of wet deposit in impacting the
AC over eastern China exists uncertainties, showing strong regional dependence. The
impact of wet deposit on the AC was examined by a sensitive experiment by turning
off the wet deposition (Fig. 11c-d). A similar anomalous AC distribution was observed
as those shown in Fig. 5, confirming that the role of wet deposit in impacting the
distribution of AC is not as important as the circulation.
**5. Summary and Discussion**
Using the simulations of GEOS-Chem model with fixed emissions, the
coordinated impacts of the previous autumn NAO and simultaneous ENSO on the
boreal winter AC over eastern China are investigated. The results present that both the
NAO and ENSO show asymmetry impacts on the boreal winter AC over eastern China,
i.e., the NAO manifests negative impacts over central China during its negative phase
and the ENSO positively impacts the AC over south China significantly during its warm
events. Consequently, the possible impacts of two cases were investigated to ascertain
the role of the NAO and ENSO on the distribution of AC over China. The winter 1997
had a co-occurrence of a negative NAO and an El Niño events, and winter 2002
corresponds to a negative NAO phase and neutral ENSO. For the winter 1997, obvious
enhanced AC were observed over eastern China, with a maximum approximately 30°N,

where the impacts of the NAO and El Niño overlap. For the winter 2002, there were generally increased AC over central China. These results suggest that the co-occurrence of a negative NAO and El Niño would worsen the air conditions over eastern China, and a solo negative NAO is associated with increased AC over central China.

The cross-seasonal impacts of the preceding autumn NAO on the following winter AC over China can be explained by the coupled air-sea bridge theory (Li and Ruan, 2018). The preceding negative NAO exhibits significant influences on the winds due to the adjustment of the wind to the anomalous SLP. The associated anomalous wind could affect the underlying regional SST, resulting in an anomalous SST tripole pattern over the North Atlantic. Since the North Atlantic SST exhibit strong persistence, this anomalous SST pattern could persist to the subsequent winter and inducing an anomalous AEA teleconnection wave train in the upper troposphere, with anomalous convergence over central China. Thus, central China is controlled by anomalous southeasterlies or easterlies, which weaken the climatological northwesterlies and induce increased AC over central China. In contrast, the occurrence of El Niño is linked to warm SST anomalies over tropical eastern Pacific, by which the Rossby wave activity would be altered (Wang et al., 2001; Feng and Li, 2011). A northwest shift of the WPSH is seen during the winter of an El Niño event, associated with southwesterlies anomalies over south China during the winter of 1997, indicating a weakening in the climatological wind and leading to enhanced AC over south China. Therefore, the high level of AC over eastern China during the winter 1997 results from the combined role

of the NAO and El Niño, and the high concentrations over central China in the winter of 2002 are attributed to the NAO.

The possible reason for the asymmetric influence of the NAO on the AC was further explored. When the autumn NAO is in the positive polarity, for example, two positive cases of 1986 and 1992, the associated underlying SST anomalies (figure not shown), particularly the tripole SST pattern, are not as evident as those shown in the negative NAO. This result may provide a possible explanation for the asymmetric relationship existed in the different phases of the NAO and AC, and implies the complexity of the atmosphere-ocean feedback in the North Atlantic. This merits further exploration related to why the linkage between the NAO and underlying SST is nonlinear, and what process is responsible for their nonlinear relationship.

As noted above, the influence of the NAO on the AC only manifests during its negative phase, and the impact of the ENSO is only significant during its warm events. However, the relationship between the previous autumn and following winter ENSO is insignificant, thus it is of interest to establish the nonlinear relationship among them and investigate why there is strong asymmetry in the relationships. Zhang et al. (2015, 2019) explored the complex linkage between the boreal winter NAO and ENSO with the former lagged for one month, indicating that the nonlinear relationship of the NAO and ENSO is modulated by the interdecadal variation in the Atlantic Multi-Decadal Oscillation. In addition, Wu et al. (2009) have illustrated the coordinated impacts of the NAO and ENSO in modulating the interannual variation of the EASM; however, it has not been shown to determine the AC yet. Therefore, it is of interest to further explore

whether the NAO and ENSO affect the AC over China in other seasons, as well as the
process involved. Furthermore, the present work is based on model simulations and due
to the limitations of the model simulations, only the interannual variations are
considered. As both NAO and ENSO show strong interdecadal variations, for a longer
period, i.e., 1850-2017 (figure not shown), the NAO during period 1986-2006 is
generally located in the positive phase, whereas in the negative phase during period
1955-1970, therefore, it is important to determine the interdecadal modulation of the
NAO on the distribution of AC.
Moreover, the role of rainfall in influencing the AC shows uncertainties, i.e., a
positive effect over south China but not for central China. This result is similar with
that of Wu (2014), showing the impact of wet deposit on the AC shows regional and
seasonal dependence. This is may due to the fact that the climatological winter rainfall
over central China is much less than that over south China (figure not shown). In
addition, the meteorological backgrounds of south China and central China are different,
baroclinic over central China and barotropic over south China (Fig. 9 vs. 10), indicating
the importance of climatology background in impacting the spatial distribution of AC.
In addition, both the NAO and ENSO show significant correlations with AC over
northwest China (Fig. 4); however, the interannual variation (Fig. 2) and anomalies (Fig.
5) in AC over those regions are relatively small. Therefore, the AC variation over those
regions are not discussed.
Finally, the role of NAO and El Niño on the AC during boreal winter was
investigated based on GEOS-Chem simulations. The coordinated role of the NAO and
El Niño in affecting the distribution of AC over eastern China is highlighted by
comparing this effect with the solo role of the NAO. The result indicates that the
influence of meteorological factors impacting AC is complicated. Future work will
investigate the combined role of tropical and extratropical signals on seasonal AC to
better understand the variation across seasons and to determine the possible
contribution of natural variability to the current aerosol loading over China.

*Author contribution*

J. F., J. L., and H. L. designed the research. J. F. and J. Z. performed the data analysis and simulations. J. F. led the writing and prepared all figures. All the authors discussed the results and commented on the manuscript.

*Data availability*

The HadISST dataset is available online at http://www.metoffice.gov.uk/hadobs/hadisst/data/download.html. The NCEP/NCAR reanalyses is available at http://www.esrl.noaa.gov/psd/data/gridded/. The model output used in the figures of this study is available at Zenodo (https://doi.org/10.5281/zenodo.3247326).

*Acknowledgement*

This work was jointly supported by the National Natural Science Foundation of China (41790474, 41705131, and 41530424).

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

**Figure Captions:**

**Figure 1**. (a) The time series of the Niño3 index based on the GEOS-4 input skin temperature data for 1986-2006 (°C). (b) is similar to (a) but is based on the HadISST. (c) The time series of the NAO index based on the GEOS-4 input sea level pressure. (d) is similar to (c) but is based on the NCEP/NCAR reanalysis.

**Figure 2**. The standard deviation of the simulated (a) surface layer $PM_{2.5}$ concentrations ($\mu g \cdot m^{-3}$) and (b) column burdens of $PM_{2.5}$ ($mg \cdot m^{-2}$) during boreal winter averaged from 1986 to 2006. (c) The horizontal distribution of boreal winter climatological mean wind at 850 hPa ($m \cdot s^{-1}$), shaded indicates the Tibetan Plateau.

**Figure 3**. (a) The spatial distribution of the correlation coefficients between surface layer $PM_{2.5}$ concentrations and the Niño3 index. (b) As in (a), but for the correlations with the NAOI. Color shading indicates a significant correlation at the 0.1 level (0.37 is the critical value for significance at the 0.1 level).

**Figure 4**. Spatial distribution of the correlation coefficients between (a) positive and (b) negative Niño3 index values and surface-layer $PM_{2.5}$ concentrations. (c)-(d) as in (a)-(b), but for the NAOI. Color shading indicates a significant correlation, (0.35 and 0.45 are the critical value for significance at the 0.2 and 0.1 level, respectively).

**Figure 5**. The spatial distribution of the simulated anomalous (left panel) surface layer $PM_{2.5}$ concentrations ($\mu g \cdot m^{-3}$) and (right panel) column burdens of $PM_{2.5}$ ($mg \cdot m^{-2}$) during the boreal winters of 1997 (upper) and 2002 (below).

**Figure 6**. The pressure–latitude distribution of zonally averaged $PM_{2.5}$ anomalies over 105°–120°E during the winters of (a)1997 and 2002 ($\mu g \cdot m^{-3}$).

**Figure 7**. The horizontal distribution of surface wind (m·s$^{-1}$) and surface level pressure
(hPa) based on the assimilated meteorological data during the autumns of (a) 1997
and (b) 2002.
**Figure 8**. The horizontal distribution of skin temperature anomalies (°C) based on the
assimilated meteorological data during the (a) autumn and (b) winter of 1997. (c)-
(d) As in (a)-(b), but during 2002.
**Figure 9**. Horizontal distribution of the divergence ($10^{-5}s^{-1}$) at 300 hPa during the
winters of (a) 1997 and (b) 2002. The crosses denote the centers of action of the
AEA pattern.
**Figure 10**. Horizontal distribution of 850 hPa wind anomalies (vectors; m s$^{-1}$) and
divergence (shading; $10^{-5}s^{-1}$) at 700 hPa during the winters of (a) 1997 and (b)

723    2002.

**Figure 11**. The spatial distribution of the vertically integrated wet deposition flux
anomalies during the winters of (a) 1997 and (b) 2002. (c)-(d), As in (a)-(b), but
for the anomalous distribution of aerosol concentrations when the wet deposit is
turned off.

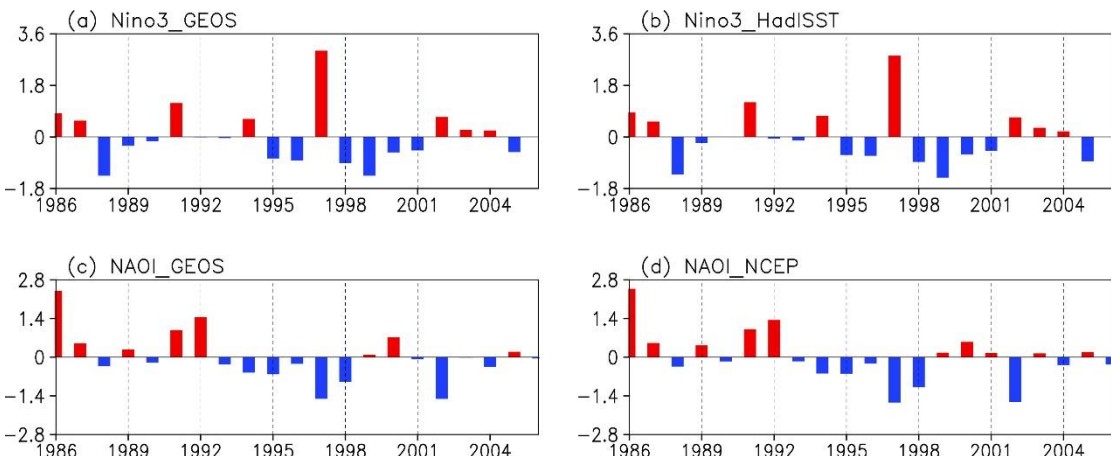


**Figure 1**. (a) The time series of the Niño3 index based on the GEOS-4 input skin
temperature data for 1986-2006 (°C). (b) is similar to (a) but is based on the HadISST.
(c) The time series of the NAO index based on the GEOS-4 input sea level pressure. (d)
is similar to (c) but is based on the NCEP/NCAR reanalysis.

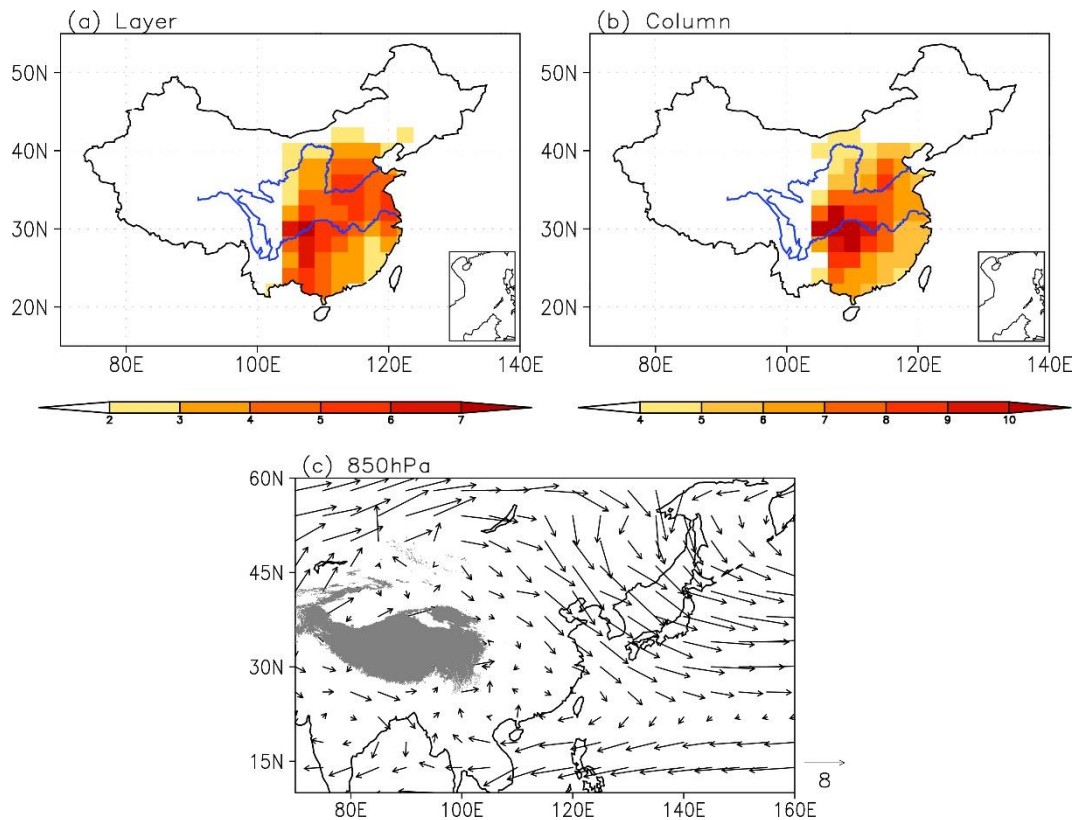


**Figure 2**. The standard deviation of the simulated (a) surface layer PM$_{2.5}$ concentrations

($\mu g \cdot m^{-3}$) and (b) column burdens of PM$_{2.5}$ (mg·m$^{-2}$) during boreal winter averaged from

1986 to 2006. (c) The horizontal distribution of boreal winter climatological mean wind

at 850 hPa (m·s$^{-1}$), shaded indicates the Tibetan Plateau.


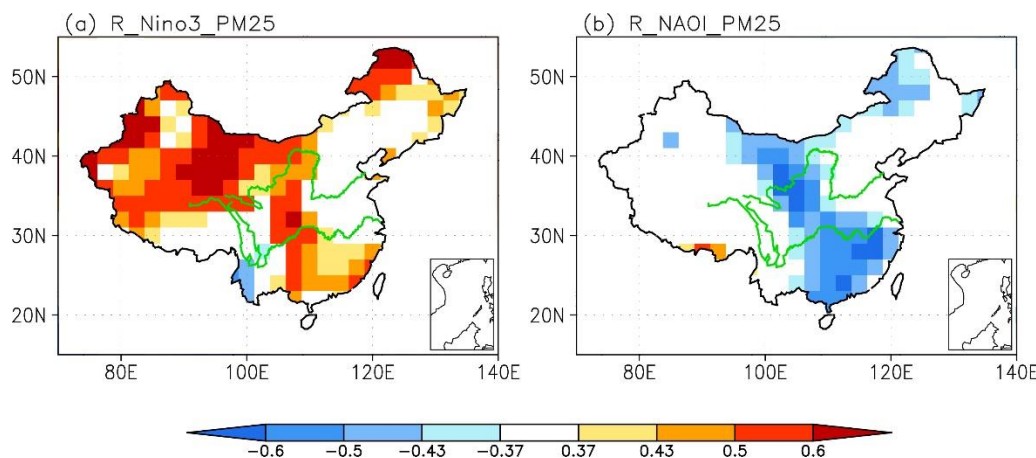


**Figure 3**. (a) The spatial distribution of the correlation coefficients between surface
layer PM$_{2.5}$ concentrations and the Niño3 index. (b) As in (a), but for the correlations
with the NAOI. Color shading indicates a significant correlation at the 0.1 level (0.37
is the critical value for significance at the 0.1 level).

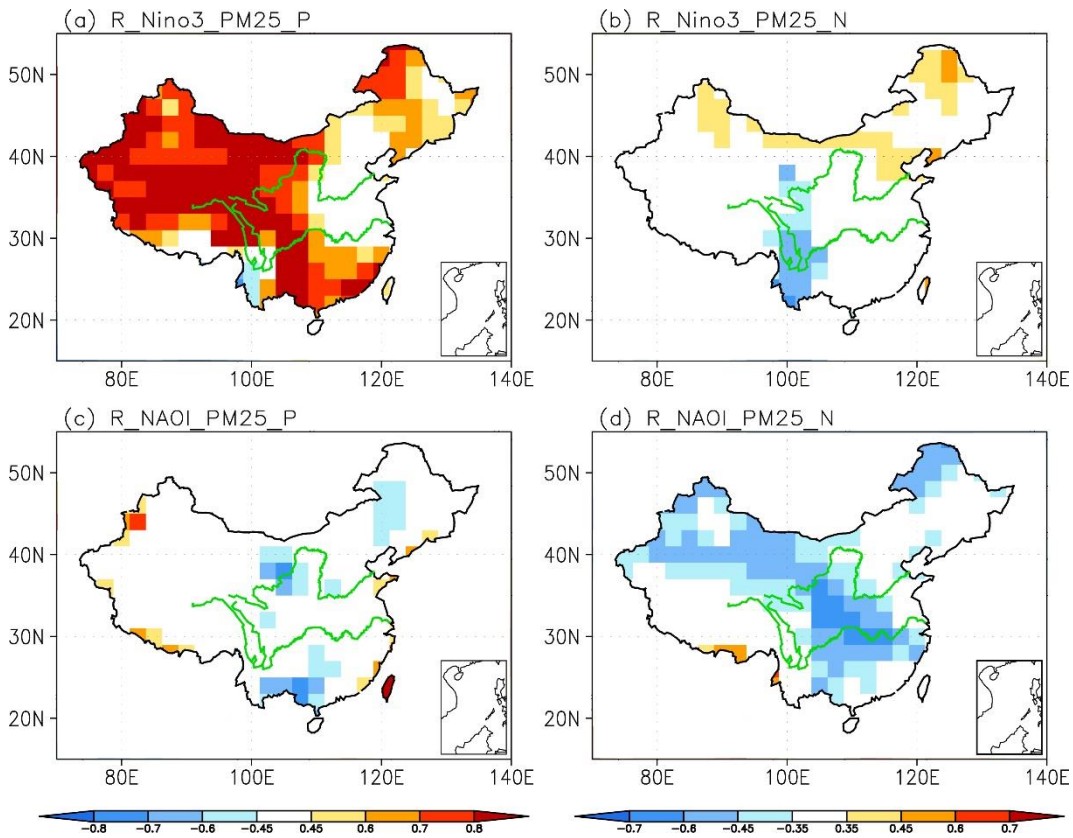


**Figure 4**. Spatial distribution of the correlation coefficients between (a) positive and (b)

negative Niño3 index values and surface-layer PM$_{2.5}$ concentrations. (c)-(d) as in (a)-

(b), but for the NAOI. Color shading indicates a significant correlation, (0.35 and 0.45

are the critical value for significance at the 0.2 and 0.1 level, respectively).

752

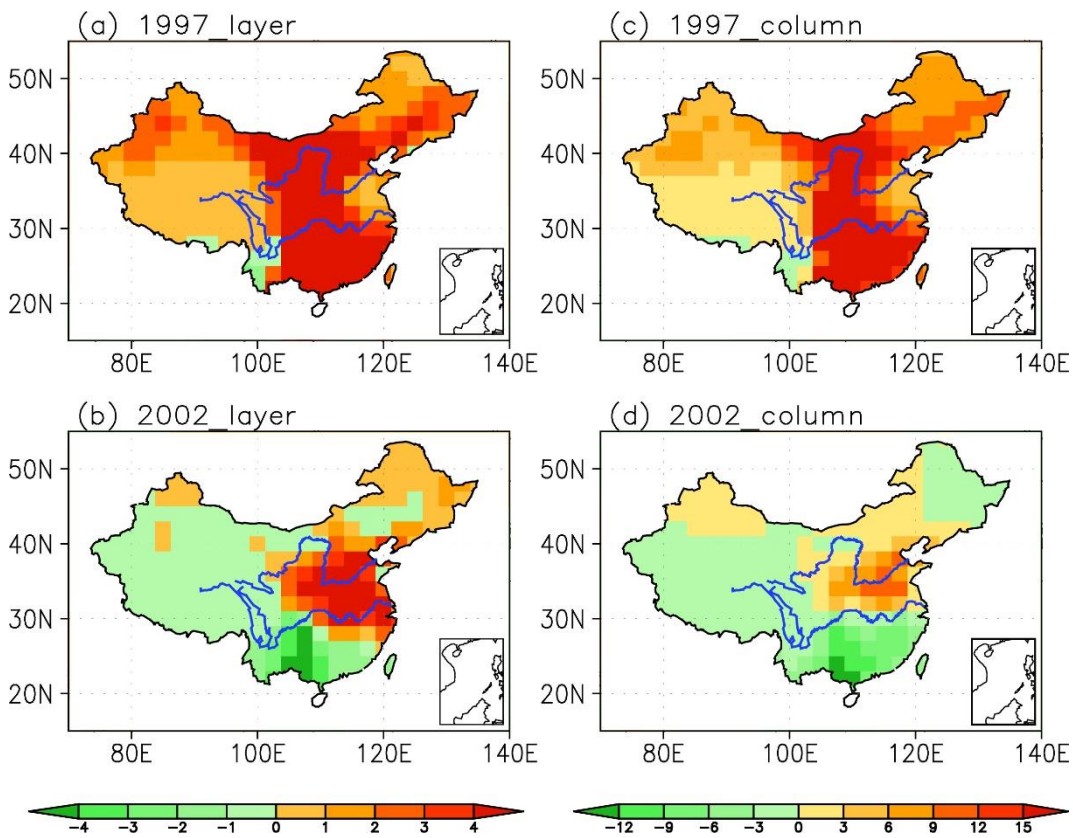

**Figure 5**. The spatial distribution of the simulated (left panel) anomalous surface layer PM$_{2.5}$ concentrations (μg·m$^{-3}$) and (right panel) column burdens of PM$_{2.5}$ (mg·m$^{-2}$) during the boreal winters of 1997 (upper) and 2002 (below).

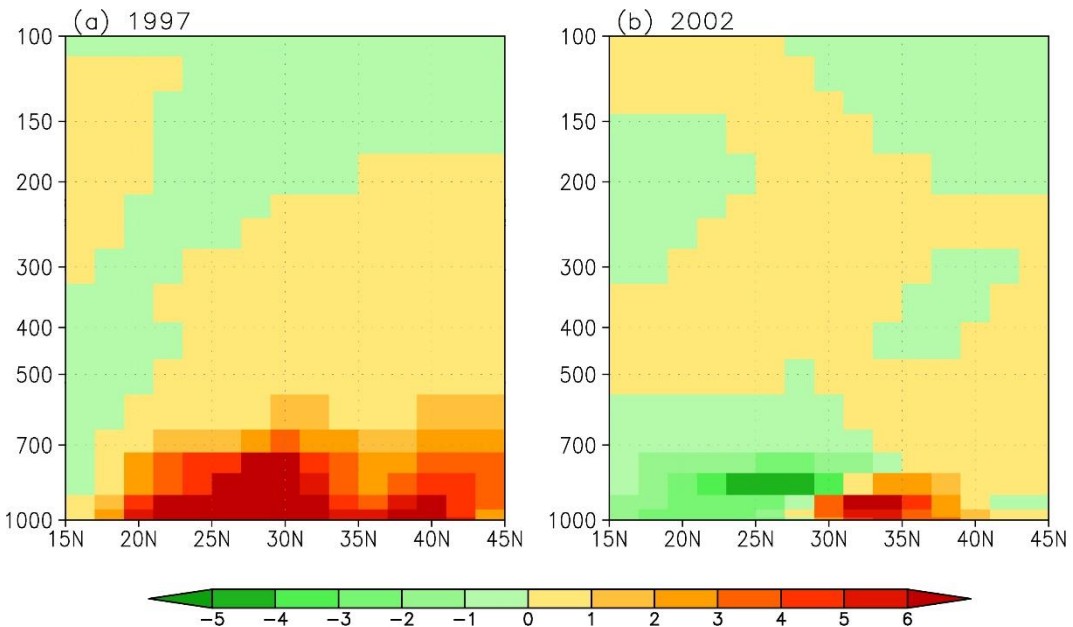


**Figure 6**. The pressure–latitude distribution of zonally averaged PM$_{2.5}$ anomalies over

105°–120°E during the winters of (a)1997 and 2002 (μg·m$^{-3}$).


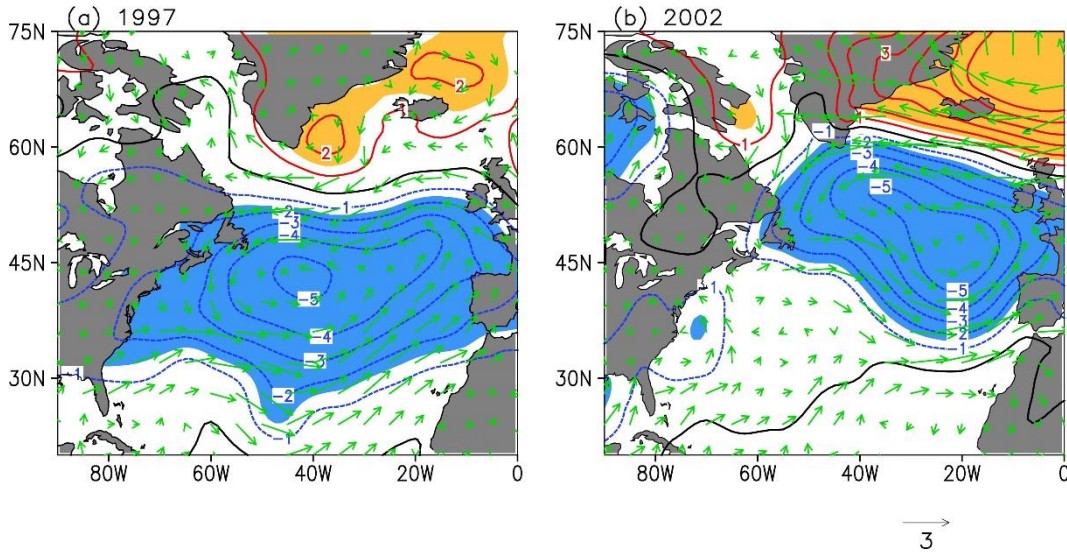


**Figure 7**. The horizontal distribution of surface wind (m·s$^{-1}$) and surface level pressure
(hPa) based on the assimilated meteorological data during the autumns of (a) 1997 and
(b) 2002.

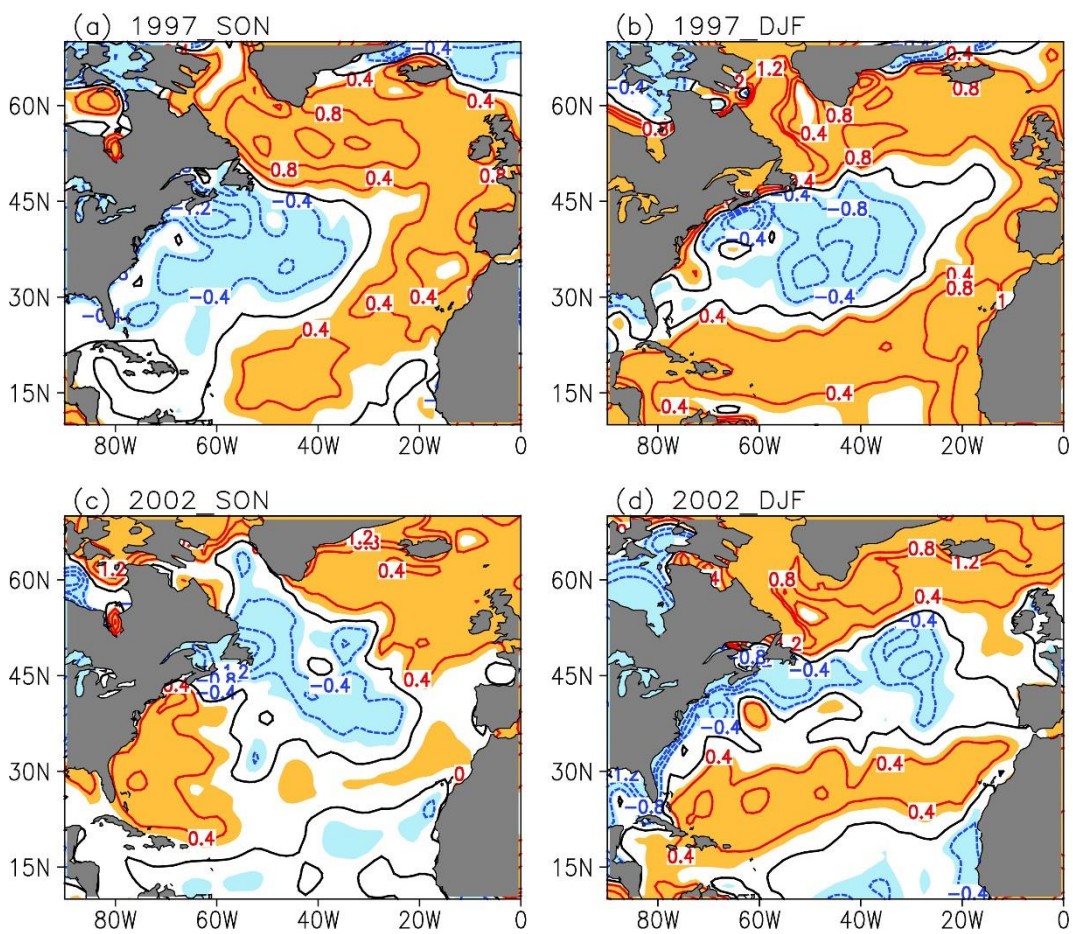

**Figure 8**. The horizontal distribution of skin temperature anomalies (°C) based on the assimilated meteorological data during the (a) autumn and (b) winter of 1997. (c)-(d) As in (a)-(b), but during 2002.

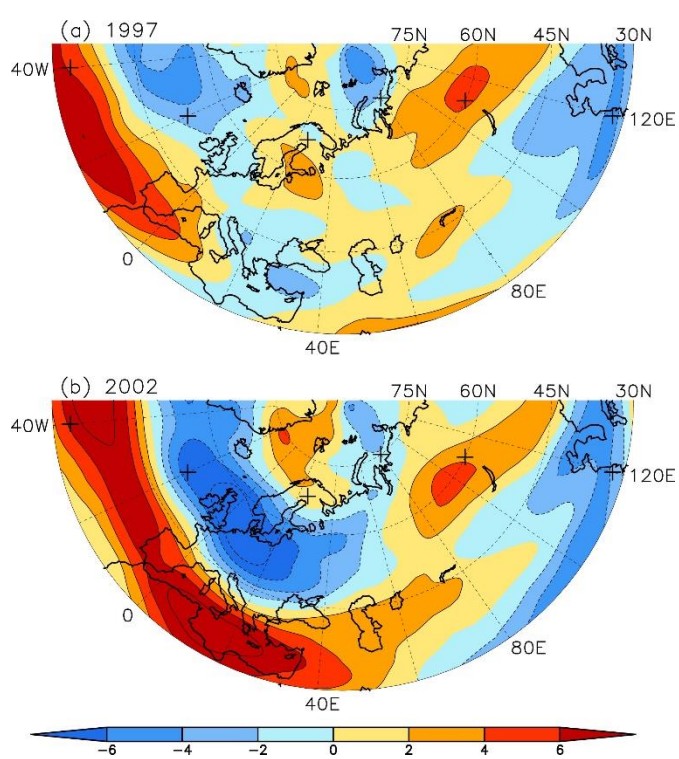

772

**Figure 9**. Horizontal distribution of the divergence ($10^{-5}s^{-1}$) at 300 hPa during the

winters of (a) 1997 and (b) 2002. The crosses denote the centers of action of the AEA

pattern.

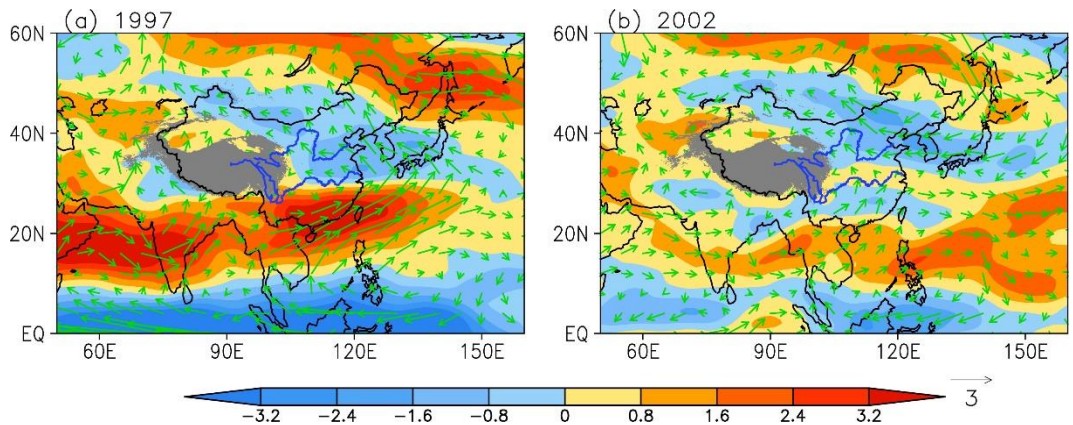

**Figure 10**. Horizontal distribution of 850 hPa wind anomalies (vectors; m s$^{-1}$) and divergence (shading; $10^{-5}$s$^{-1}$) at 700 hPa during the winters of (a) 1997 and (b) 2002.

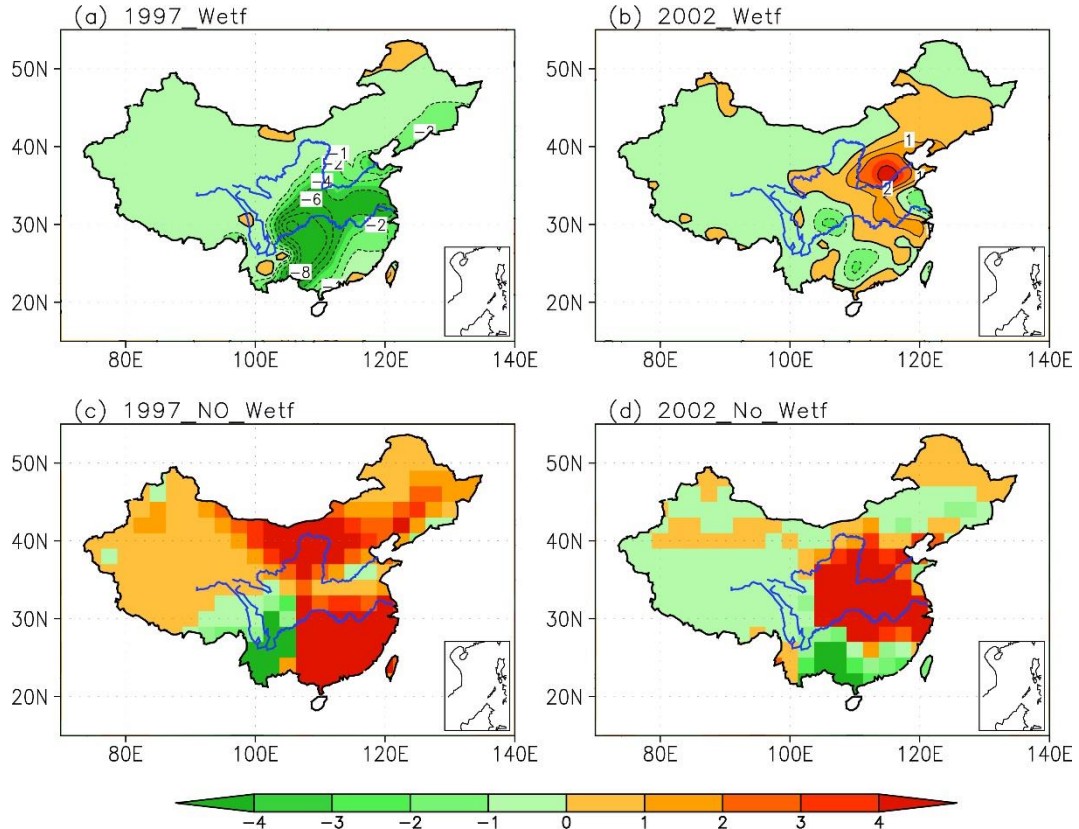

781

**Figure 11**. The spatial distribution of the vertically integrated wet deposition flux anomalies during the winters of (a) 1997 and (b) 2002. (c)-(d), As in (a)-(b), but for the anomalous distribution of aerosol concentrations when the wet deposit is turned off.