# Peer review of "Simulated coordinated impacts of the previous autumn NAO and winter El Niño on the winter aerosol concentrations over eastern China"

_Atmospheric Chemistry and Physics, 2019_

## Referee Comment (RC1) · Anonymous Referee #1 · 21 Mar 2019

The manuscript investigated the impact of atmospheric circulation (NAO and ENSO) on the high aerosol concentration in Eastern China was investigated by using simulations of GOES-4, which can exclude the influence of emission. They found that the asymmetric impact of NAO and ENSO on the AC over central and eastern China, and further discussed the physical mechanism induced the circulation anomalies associated with NAO- and El Nino.

In general, I found the paper appropriate for ACP. However, it need to be major revised before accepted this paper for publication in ACP with addressing those comments listed below:

Major Comments: 1. The manuscript focus on the winter high aerosol concentration and its interannual variation associated with NAO and El Nino, so I think, it is better to point out the seasonal information and time scale of variation in the title to avoid misleading, such as "Simulated coordinated impacts of the previous autumn NAO and winter El Nino on the interannual variation of winter aerosol concentrations over eastern China.", Certainly, authors can give a better title than this.

2. the introduction mentions that "NAO exhibits significant cross-seasonal impacts on the East Asian climate, ... boreal spring NAO influenced the subsequent intensity of EASM". However, the manuscript investigates the influence of autumn NAO on winter climate, so I think it is better to providing some references to explain why we should investigate the impact of autumn NAO on winter climate.

3. the time scale of NAO and ENSO are different, the impact of NAO is mainly in decadal time scale, ENSO is mainly in interannual time scale. The time scale should be clarifying clear when authors get the conclusion.

4. as the authors said, although the NOA index are close in 1997 and 2002 (-1.507 in 1997 and -1.510 in 2002), the precise location of anomalous SLP is different, so the difference of AC distribution in 1997 and 2002 (fig. 5a,b) may be caused by the difference of locations of anomalous SLP pattern associated with NOA, the author should give some explain before investigate the impact of NAO &EI Nino and solo NAO on the AC.

Minor Comments: 1. Line 225 and Fig. 4, the significant level is 0.2 level, which is different with fig. 3 (0.1 level) and it is too lower in the statistical significance. Suggest to use consistent significant level (like 0.1 level).

2. Line 232, like author mentioned in Line406-408, author should point out that "the ENSO affects the distribution of AC in south China and northwest China." Northwest China is not discussed but should be noted based on the figure.

3. Line 248 and the legend in Fig. 5, "column AC anomalies " in the maintext, however, the legend of Fig. 5 did not point out "anomalies", which is right? Maybe the maintext is right.

4. Line 281: "negative SST " -> "negative SST anomaly"

5. Line 300-301, "Under the influence of the anomalous downstream teleconnection, north China is influenced by convergence anomalies, with the center positioned over central China (Fig. 9)." The Fig. 9 can not fully support this sentence, maybe due to the missing lon information in the Fig. 9 or the country boundaries. I suggest to make the fig. 9 more clear.

6. "convergence" in Line 308 and "anomalous divergence" in Line 313, which is contrary to the Fig. 10. Generally, the negative values of divergence indicate convergence, positive values indicate divergence. Therefore, Line 308 said "south China was influenced by an evident anomalous convergence at the lower troposphere.." however, I see the positive values (orange color) of divergence in Fig. 10a. please check it.

---

## Referee Comment (RC2) · Anonymous Referee #2 · 21 Mar 2019

The manuscript presents analysis of the impacts of NAO and El Niño on the anthropogenic aerosols in China. It uses mostly GEOS-Chem model simulations driven by GEOS-4 reanalysis. Understanding the changes in aerosols is a relevant topic for improving our knowledge of relationship between natural cycle and aerosols. Model simulation show the circulation anomalies during the co-occurrence events of negative NAO and El Niño, and therefore influence on aerosol concentrations over eastern China. However, a sole negative NAO is linked with anomalous aerosols over central China. Overall the manuscript is well written and clear, the figures are also appropriate and clear. After addressing the following minor concerns, I suggest publishing this work.

[Figure]

1. I suggest that the authors could also select more sample size (negative NAO + El Niño and El Niño events) from reanalysis data in a longer time, i.e., 1979-2016, and compare the distribution of wind anomalies. 2. Data part. In the whole study, the authors used mostly the model data, even in analyzing the atmospheric circulation, why? And what are the differences between model data and reanalysis data? Can their differences influence the results of the research? 3. Figure 11, discuss the contribution from wet deposition, I think the limited role of wet deposit on the aerosol concentrations over central China is partly due to the small amount of rainfall during winter. However, the winter rainfall amount over south China is greater than that over central China. The author should further examine this point. 4. Finally, why the impacts of positive NAO on the aerosol concentrations are insignificant, the authors should shed more light on this issue. The corresponding variations in the underlying thermal and dynamical process should be included to give a full understanding. 5. The related reference the authors might be interested in: Li, X., Z. Wu and Y. Li, 2019: A link of China warming hiatus with the winter sea ice loss in Barents–Kara Seas. Clim Dyn., DOI:10.1007/s00382-019-04645-z. Wu, J. and Z. Wu, 2018: Interdecadal change of the spring NAO impact on the summer Pamir-Tienshan Snow Cover. Int.J. Climatol., DOI: 10.1002/joc.5831. Wu, Z., X. Li, Y. Li and Y. Li, 2016: Potential Influence of Arctic Sea Ice to the Inter-annual Variations of East Asian Spring Precipitation. J. Clim., 29, 2797-2813. Wu, Z., J. Li, Z. Jiang and J. He, 2011: Predictable climate dynamics of abnormal East Asian winter monsoon: once-in-a-century snowstorms in 2007/2008 winter. Climate Dyn., 37, 1661-1669. Lyu, M., Z. Wu, X. Shi and M. Wen, 2019: Distinct effects of the MJO and the NAO on cold wave amplitude over China. Quart. J. Roy. Meteor. Soc., DOI: 10.1002/qj.3516. Zhang, P., B. Wang and Z. Wu, 2019: Weak El Niño and Winter Climate in the mid-high latitude Eurasia. J. Climate, 32, 402ïĂ▪421. Zhang, P., Z. Wu and J. Li, 2019: Reexamining the relationship of La Niña and the East Asian winter monsoon. Climate Dyn., DOI: 10.1007/s00382-019-04613-7. Ye, X. and Z. Wu, 2018: Contrasting Impacts of ENSO on the Interannual Variations of Summer Runoff between the Upper and Mid-Lower Reaches of the Yangtze River. Atmosphere,

DOI: 10.3390/atmos9120478. Zhang, P., Z. Wu, and H. Chen, 2017: Interdecadal Modulation of mega-ENSO on the North Pacific Atmospheric Circulation in Winter. Atmos.ïĂăïĂ▪Ocean, 55(2), 110ïĂ▪120. Zhou, Y., and Z. Wu, 2016: Possible impacts of mega-El Niño/Southern Oscillation and Atlantic multidecadal oscillation on Eurasian heat wave frequency variability. Quart. J. Roy. Meteor. Soc., 142, 1647ïĂ▪1661. Wu, Z., and P. Zhang, 2015: Interdecadal Variability of the mega-ENSO-NAO Synchronization in Winter. Climate Dyn., 45, 1117ïĂ▪1128. Wu, Z. and H. Lin, 2012: Interdecadal Variability of the ENSO-North Atlantic Oscillation Connection in boreal summer. Quart. J. Roy. Meteor. Soc., 138, 1668ïĂ▪1675, DOI: 10.1002/qj.1889. Wu, Z., J. Li, Z. Jiang, J. He and X. Zhu, 2012: Possible effects of the North Atlantic Oscillation on the strengthening relationship between the East Asian summer monsoon and ENSO. Int. J. Climatol., 32, 794ïĂ▪800. DOI: 10.1002/joc.2309.
* * *
* * *

---

## Referee Comment (RC3) · Anonymous Referee #2 · 28 Mar 2019

I am satisfied with the reviews and have no further comments.

---

## Referee Comment (RC4) · Anonymous Referee #2 · 28 Mar 2019

I am satisfied with the reviews and have no further comments.

---

## Referee Comment (RC5) · Anonymous Referee #2 · 28 Mar 2019

I am satisfied with the reviews and have no further comments.

---

## Author Comment (AC1) · 28 Mar 2019

**Response to Comments of Reviewer A**

**Manuscript number**: acp-2019-62

**Author(s)**: Juan Feng, Jianping Li, Hong Liao, and Jianlei Zhu

**Title**: Simulated coordinated impacts of the previous autumn NAO and winter El Niño on the winter aerosol concentrations over eastern China

**General comments:**

*The manuscript investigated the impact of atmospheric circulation (NAO and ENSO) on the high aerosol concentration in Eastern China was investigated by using simulations of GOES-4, which can exclude the influence of emission. They found that the asymmetric impact of NAO and ENSO on the AC over central and eastern China, and further discussed the physical mechanism induced the circulation anomalies associated with NAO- and El Nino. In general, I found the paper appropriate for ACP. However, it need to be major revised before accepted this paper for publication in ACP with addressing those comments listed below:*

**Response:**

Thanks to the reviewer for the helpful comments and suggestions. We have revised the manuscript seriously and carefully according to the reviewer's comments and suggestions. The point-to-point responses to the comments are listed as follows.

**Major Comments:**

1. *The manuscript focus on the winter high aerosol concentration and its interannual variation associated with NAO and El Nino, so I think, it is better to point out the seasonal information and time scale of variation in the title to avoid misleading, such as "Simulated coordinated impacts of the previous autumn NAO and winter El Nino on the interannual variation of winter aerosol concentrations over eastern China.", Certainly, authors can give a better title than this.*

**Response:**

We have adopted the reviewer's comment and revised the title. Since the El Niño is mainly an interannual variability, we have omitted the interannual variation in the

suggested title.

2. *The introduction mentions that "NAO exhibits significant cross-seasonal impacts on the East Asian climate, ... boreal spring NAO influenced the subsequent intensity of EASM". However, the manuscript investigates the influence of autumn NAO on winter climate, so I think it is better to providing some references to explain why we should investigate the impact of autumn NAO on winter climate.*

**Response:**

Thanks to the reviewer for the comments. Previous studies have found that spring (summer) NAO plays important role in impacting the summer (autumn) climate over eastern China, indicating the impact of NAO on the East Asian climate is cross-seasonal. We have examined the role of previous autumn and simultaneous winter NAO on the winter aerosols over eastern China, and it is found the influences of winter NAO on the aerosols are insignificant (Figure R1). Based on the above discussions, the role of previous autumn NAO on the AC over eastern China is discussed in the present work.

[Figure]

**Figure R1**. The spatial distribution of the correlation coefficients between surface layer PM$_{2.5}$ concentrations and the winter NAOI.

3. *the time scale of NAO and ENSO are different, the impact of NAO is mainly in decadal time scale, ENSO is mainly in interannual time scale. The time scale should be clarifying clear when authors get the conclusion.*

**Response:**

The reviewer is right that the NAO exhibits strong decadal variation. For the longer period, for example, 1850-2017, strong decadal variation is observed in the NAOI (Figure R2). However, as shown the NAO in the period 1986-2006 is generally

located in the positive phase, and is characterized by strong interannual variations. We have included the above discussions into the revised manuscript.

[Figure]

**Figure R2**. The annual mean NAO index during 1850-2017.

4. *as the authors said, although the NAO index are close in 1997 and 2002 (-1.507 in 1997 and -1.510 in 2002), the precise location of anomalous SLP is different, so the difference of AC distribution in 1997 and 2002 (fig. 5a,b) may be caused by the difference of locations of anomalous SLP pattern associated with NAO, the author should give some explain before investigate the impact of NAO &El Nino and solo NAO on the AC.*

**Response:**

Thanks for the comment. From the correlation between the AC and the NAOI during its negative phases, significant negative correlations are seen over central China, indicating a negative NAO is connected with enhanced AC over the central China. However, there is no significant signal over the south China in the correlations between the positive NAOI and AC, indicating the role of positive NAO on the AC over south China is limited. Besides, enhanced AC anomalies are seen over central China in both 1997 and 2002 winters, and similar teleconnection wave train is observed in both winters, suggesting the role of NAO on the AC over central China.

Moreover, the effect of El Niño in impacting the distribution of AC is confirmed for that warm El Niño event is associated with enhanced AC over south China. The influences of El Niño on the circulation and rainfall over south China has been discussed in previous studies (e.g., Weng et al., 2007, 2009; Feng and Li, 2011; Feng et al., 2016). The above discussion provides confidence for the combined role of NAO and El Niño on the boreal AC over eastern China.

As the reviewer pointed that the locations of the anomalous pressure centers in the two negative NAO events show difference, however, it is seen that the two events bear equivalent index values, and with similar anomalous SLP amplitude, i.e., with bigger negative SLP anomalies and the maximum minus center is same. That is the pressure gradient of the two NAO negative events is similar, contributing to the similar anomalous SST pattern and teleconnection wave train as shown in the manuscript.

The above discussions indicate the combined impacts of the NAO and El Niño on the boreal winter AC over eastern China.

**Minor Comments:**

1. *Line 225 and Fig. 4, the significant level is 0.2 level, which is different with fig. 3 (0.1 level) and it is too lower in the statistical significance. Suggest to use consistent significant level (like 0.1 level).*

**Response:**

Thanks. Different significance level is shown due to that the sample in Figure 4 is less than that in Figure 3. The possible different impacts between the negative and positive phases of NAO, as well as between the warm and cold events of ENSO are discussed, whereas the whole period. In fact, the color bar 0.35 is for the significance at 0.2 level, and 0.45 is for the significance at 0.1 level, we see that the different significance level would not change the result. We have added the detailed caption into the revised manuscript.

2. *Line 232, like author mentioned in Line406-408, author should point out that "the ENSO affects the distribution of AC in south China and northwest China." Northwest China is not discussed but should be noted based on the figure.*

**Response:**

We have revised the relevant description.

3. *Line 248 and the legend in Fig. 5, "column AC anomalies " in the maintext, however, the legend of Fig. 5 did not point out "anomalies", which is right? Maybe the main text is right.*

**Response:**

The reviewer is right, it is for the anomalous aerosol concentrations, and we have revised the relevant description.

4. *Line 281: "negative SST " -> "negative SST anomaly"*

**Response:**

Yes, done.

5. *Line 300-301, "Under the influence of the anomalous downstream teleconnection, north China is influenced by convergence anomalies, with the center positioned over central China (Fig. 9)." The Fig. 9 can not fully support this sentence, maybe due to the missing lon information in the Fig. 9 or the country boundaries. I suggest to make the fig. 9 more clear.*

**Response:**

We have adopted the reviewer's comment and added the longitude and latitude in to the revised Figure 9.

6. *"convergence" in Line 308 and "anomalous divergence" in Line 313, which is contrary to the Fig. 10. Generally, the negative values of divergence indicate convergence, positive values indicate divergence. Therefore, Line 308 said "south China was influenced by an evident anomalous convergence at the lower troposphere." however, I see the positive values (orange color) of divergence in Fig. 10a. please check it.*

**Response:**

Sorry for the typo, the reviewer is right. In winter 1997, there are anomalous divergence over the southeastern coastal regions of China, associated with anticyclonic circulation anomalies. We have revised the description.

---

## Author Comment (AC2) · 28 Mar 2019

**Response to Comments of Reviewer B**

**Manuscript number**: acp-2019-62

**Author(s)**: Juan Feng, Jianping Li, Hong Liao, and Jianlei Zhu

**Title**: Simulated coordinated impacts of the previous autumn NAO and winter El Niño on the winter aerosol concentrations over eastern China

**General comments:**

*The manuscript presents analysis of the impacts of NAO and El Niño on the anthropogenic aerosols in China. It uses mostly GEOS-Chem model simulations driven by GEOS-4 reanalysis. Understanding the changes in aerosols is a relevant topic for improving our knowledge of relationship between natural cycle and aerosols. Model simulation show the circulation anomalies during the co-occurrence events of negative NAO and El Niño, and therefore influence on aerosol concentrations over eastern China. However, a sole negative NAO is linked with anomalous aerosols over central China. Overall the manuscript is well written and clear, the figures are also appropriate and clear. After addressing the following minor concerns, I suggest publishing this work.*

**Response:**

   Thanks to the reviewer for the helpful comments and suggestions. We have revised the manuscript seriously and carefully according to the reviewer's comments and suggestions. The point-to-point responses to the comments are listed as follows.

**Comments:**

1. *I suggest that the authors could also select more sample size (negative NAO + El Niño and El Niño events) from reanalysis data in a longer time, i.e., 1979-2016, and compare the distribution of wind anomalies.*

**Response:**

   Thanks for the comment. We have adopted the reviewer's comment by examining the temporal variation of autumn NAO and winter El Niño. Except the cases in the manuscript, there is only one well defined negative NAO event, i.e., 2010, and one El Niño event, i.e., 2015. However, the occurrence of the negative NAO is overlapped

with a La Niña event, and the El Niño event 2015 is along with a neutral NAO event. For the El Niño event 1982, it is along with a positive NAO. That is there is no other proper cases (negative NAO + El Niño) as shown in the manuscript during period 1979-2016.

[Figure]

**Figure R1.** (a) The time series of the Niño3 index based on the HadISST during period 1979-2016. (b) The time series of the NAO index based on the NCEP/NCAR reanalysis during period 1979-2016.

2. *Data part. In the whole study, the authors used mostly the model data, even in analyzing the atmospheric circulation, why? And what are the differences between model data and reanalysis data? Can their differences influence the results of the research?*

**Response:**

    Thanks to the reviewer for the helpful comments. We would like to clarify the reliability of the datasets used by the following two points:

1) We have shown in the manuscript, the input surface skin temperature of GEOS-Chem is highly correlated with the widely used SST dataset, i.e., HadISST. And the NAOI based on GEOS-Chem is closely correlated with the NAOI based on the NCEP/NCAR reanalysis.

2) The input meteorological fields (GEOS-4), such as winds, temperature, humidity, have been evaluated in Zhu et al. (2012) and Feng et al. (2016), and their result suggested the GEOS-Chem input meteorological fields are highly

consistent with the NCEP/NCAR reanalysis. Besides, the spatial distribution of winds anomalies at 850 hPa during two events, i.e., 1997 and 2002, based on the GEOS-Chem input meteorological fields and those from NCEP/NCAR are computed as shown in Figure R2. We see that the winds show similar spatial structures, implying high consistency between the model meteorological fields and reanalysis. The above results provide confidence for the reliability of the meteorological fields of GEOS-Chem model.

[Figure]

**Figure R2**. The horizontal distribution of wind anomalies at 850 hPa during 1997 and 2002 winters based on the GEOS-Chem input meteorological winds (left panel) and NCEP/NCAR reanalysis (right panel).

The above discussions provide confidence for employing the GEOS-Chem to explore the influences of climatic events on aerosol concentrations, and it is proved to be a useful tool to understand the impacts of climatic event on aerosol concentrations without enough observations.

3. *Figure 11, discuss the contribution from wet deposition, I think the limited role of wet deposit on the aerosol concentrations over central China is partly due to the small amount of rainfall during winter. However, the winter rainfall amount over south China is greater than that over central China. The author should further examine this point.*

**Response:**

We have adopted the reviewer's comment by further examine the climatological winter rainfall distribution over China. The reviewer is right, the amount of winter rainfall over central China is much less than over south China, indicating a less important role of wet deposit on the boreal winter AC over central China than over south China. We have included this point into the revised manuscript.

[Figure]

**Figure R3**. The distribution of climatological boreal winter rainfall.

4. *Finally, why the impacts of positive NAO on the aerosol concentrations are insignificant, the authors should shed more light on this issue. The corresponding variations in the underlying thermal and dynamical process should be included to give a full understanding.*

**Response:**

We have adopted the reviewer's comment by further examining the situation during the positive NAO events. During period 1986-2006, there are two well-defined NAO positive events, i.e., 1986 and 1992. The anomalous SST pattern during the two events are shown below. It is seen the anomalous SST tripole pattern is not observed during the positive NAO events, indicating that the air-sea feedback during the positive and negative NAO events is different. Therefore, due to the different anomalous SST pattern, the teleconnection wave train during the positive NAO events are different, without significant impacts on the circulation over eastern China. We have included this point into the revised manuscript.

[Figure]

**Figure R4**. The horizontal distribution of skin temperature anomalies (°C) based on the assimilated meteorological data during the (a) autumn and (b) winter of 1986. (c)-(d) As in (a)-(b), but during 1992.

5. *The related reference the authors might be interested in:*

    *Li, X., Z. Wu and Y. Li, 2019: A link of China warming hiatus with the winter sea ice loss in Barents–Kara Seas. Clim Dyn., DOI:10.1007/s00382-019-04645-z.*

    *Wu, J. and Z. Wu, 2018: Interdecadal change of the spring NAO impact on the summer Pamir-Tienshan Snow Cover. Int.J. Climatol., DOI: 10.1002/joc.5831.*

    *Wu, Z., X. Li, Y. Li and Y. Li, 2016: Potential Influence of Arctic Sea Ice to the Inter-annual Variations of East Asian Spring Precipitation. J. Clim., 29, 2797-2813.*

    *Wu, Z., J. Li, Z. Jiang and J. He, 2011: Predictable climate dynamics of abnormal East Asian winter monsoon: once-in-a-century snowstorms in 2007/2008 winter. Climate Dyn., 37, 1661-1669.*

    *Lyu, M., Z. Wu, X. Shi and M. Wen, 2019: Distinct effects of the MJO and the NAO on cold wave amplitude over China. Quart. J. Roy. Meteor. Soc., DOI: 10.1002/qj.3516.*

    *Zhang, P., B. Wang and Z. Wu, 2019: Weak El Niňo and Winter Climate in the mid-high latitude Eurasia. J. Climate, 32, 402-421.*

    *Zhang, P., Z. Wu and J. Li, 2019: Reexamining the relationship of La Niña and the East Asian winter monsoon. Climate Dyn., DOI: 10.1007/s00382-019-04613-7.*

    *Ye, X. and Z. Wu, 2018: Contrasting Impacts of ENSO on the Interannual Variations*

*of Summer Runoff between the Upper and Mid-Lower Reaches of the Yangtze River. Atmosphere, DOI: 10.3390/atmos9120478.*

*Zhang, P., Z. Wu, and H. Chen, 2017: Interdecadal Modulation of mega-ENSO on the North Pacific Atmospheric Circulation in Winter. Atmos. Ocean, 55(2), 110-120.*

*Zhou, Y., and Z. Wu, 2016: Possible impacts of mega-El Niño/Southern Oscillation and Atlantic multidecadal oscillation on Eurasian heat wave frequency variability. Quart. J. Roy. Meteor. Soc., 142, 1647-1661.*

*Wu, Z., and P. Zhang, 2015: Interdecadal Variability of the mega-ENSO-NAO Synchronization in Winter. Climate Dyn., 45, 1117ï˘A 1128.*

*Wu, Z. and H. Lin, 2012: Interdecadal Variability of the ENSO-North Atlantic Oscillation Connection in boreal summer. Quart. J. Roy. Meteor. Soc., 138, 1668-1675, DOI: 10.1002/qj.1889.*

*Wu, Z., J. Li, Z. Jiang, J. He and X. Zhu, 2012: Possible effects of the North Atlantic Oscillation on the strengthening relationship between the East Asian summer monsoon and ENSO. Int. J. Climatol., 32, 794-800. DOI: 10.1002/joc.2309.*

**Response:**

We have updated the references and included the relevant references into the revised manuscript. More details are seen in the revised manuscript.

---

## Author Comment (AC3) · 28 Mar 2019

The comment was uploaded in the form of a supplement:
https://www.atmos-chem-phys-discuss.net/acp-2019-62/acp-2019-62-AC3-supplement.pdf

---

## Author Comment (AC4) · 28 Mar 2019

The comment was uploaded in the form of a supplement:
https://www.atmos-chem-phys-discuss.net/acp-2019-62/acp-2019-62-AC4-supplement.pdf

---

## Author Comment (AC5) · 28 Mar 2019

The comment was uploaded in the form of a supplement:
https://www.atmos-chem-phys-discuss.net/acp-2019-62/acp-2019-62-AC5-supplement.pdf

---

## Referee Comment (RC6) · Anonymous Referee #1 · 29 Mar 2019

I am satisfied with the reviews and have no further comments.